# Evaluating the Causal Association between Inflammatory Bowel Disease and Risk of Atherosclerotic Cardiovascular Disease: Univariable and Multivariable Mendelian Randomization Study

**DOI:** 10.3390/biomedicines11092543

**Published:** 2023-09-15

**Authors:** Baike Liu, Zijian Qin, Zhaolun Cai, Zheran Liu, Yun-Lin Chen, Xiaonan Yin, Yuan Yin, Xingchen Peng, Bo Zhang

**Affiliations:** 1Gastric Cancer Center, Department of General Surgery, West China Hospital, Sichuan University, Chengdu 610041, China; 2State Key Laboratory of Biotherapy, Sichuan University, Chengdu 610041, China; 3Department of Biotherapy and National Clinical Research Center for Geriatrics, Cancer Center, West China Hospital, Sichuan University, Chengdu 610041, China; 4Department of Cardiology, The 2nd Affiliated Hospital of Chongqing Medical University, Chongqing 400010, China

**Keywords:** inflammatory bowel disease, Crohn’s disease, ulcerative colitis, atherosclerotic cardiovascular disease, coronary artery disease, ischemic stroke, Mendelian randomization study

## Abstract

Background: Observational studies suggested that inflammatory bowel disease (IBD) (i.e., Crohn’s disease [CD] and ulcerative colitis [UC]) is associated with an increased risk of atherosclerotic cardiovascular disease (ASCVD), including coronary artery disease (CAD) and ischemic stroke. However, it is still unclear whether the observed associations causally exist. Thus, we aim to examine the potential effect of IBD, CD, and UC on the risk of CAD and ischemic stroke, using a two-sample Mendelian randomization (MR) study. Methods: Genetic instruments for IBD, CD, and UC were retrieved from the latest published genome-wide association studies (GWASs) of European ancestry. GWAS summary data for instrument–outcome associations were gathered from four independent resources: CARDIoGRAMplusC4D Consortium, MEGASTROKE consortium, FinnGen, and UK Biobank. The inverse variance weighted (IVW) method and multiple pleiotropy-robust approaches were conducted and, subsequently, combined in a fixed-effect meta-analysis. Moreover, multivariable MR (MVMR) analysis was conducted to adjust for potential influencing instrumental variables. Results: The IVW method revealed no causal effect of IBD on the risk of CAD (overall IBD on CAD: OR 1.003, 95%CI 0.982 to 1.025; CD on CAD: OR 0.997, 95%CI 0.978 to 1.016; UC on CAD: OR 0.986, 95%CI 0.963 to 1.010) or the risk of ischemic stroke (overall IBD on ischemic stroke: OR 0.994, 95%CI 0.970 to 1.018; CD on ischemic stroke: OR 0.996, 95%CI 0.979 to 1.014; UC on ischemic stroke: OR 0.999, 95%CI 0.978 to 1.020). The results of the meta-analysis and MVMR remained consistent. Conclusion: Our MR analysis does not support a causal effect of IBD on CAD and ischemic stroke, and previous results from observational studies might be biased through uncontrolled confoundings (such as IBD-specific medications and detection bias, etc.) that warrant further research.

## 1. Introduction

Atherosclerotic cardiovascular disease (ASCVD), including its two major subtypes, coronary artery disease (CAD) and ischemic stroke, is the leading cause of death worldwide, with mortalities of more than half a million in the United States in 2018, and approximately 2.4 million in China in 2016 [1,2]. In recent decades, it has been suggested that chronic systemic inflammation is associated with the pathogenesis of atherosclerosis [3]. Inflammatory bowel disease (IBD), encompassing Crohn’s disease (CD) and ulcerative colitis (UC), is mainly characterized by its chronic inflammatory manifestation in the intestinal tract [4]. Thus, the potential interplay between IBD and ASCVD has become a study area of interest among researchers [5]. An accumulation of evidence from observational studies has suggested an association between ASCVD and IBD. A Danish cohort study showed a significantly increased risk of ischemic heart disease incidents in patients with IBD [6]. A similar association has also been observed in a retrospective multicenter cohort study in China recently [7]. In addition, the potential association was further substantiated via some other observational studies and meta-analyses [8,9,10,11].

However, the previously discovered association between ASCVD and IBD was based on observational cohort studies, which are prone to potential confounding factors and unmeasured biases, such as medications [12]. More importantly, even with the robust statistical correlation that has been found in observational studies, the underlying causal relationship is still hardly proven [13]. Over the last decade, genome-wide association studies (GWASs) have emerged as a vital approach toward carrying out extensive analysis on genetic data originating from a large number of participants, with a prime focus on recognizing genetic variants strongly linked to a particular trait or disease. This method encompasses the comparison of the genetic profiles of individuals who may or may not be afflicted with the concerned trait or disease. Consequently, the identification of genetic variants that either escalate the risk, or provide a protective role against the given trait or disease, is attained. Recently, the largest GWAS of CAD [14], ischemic stroke [15], and IBD [16] provides a valuable opportunity to explore the potential causal association between IBD and ASCVD, through two-sample mendelian randomization (MR) analysis.

Briefly, MR serves as a novel approach toward inferring causal correlations between modifiable risk factors and health outcomes through using genetic variants as instrumental variables [13]. The advantage of MR is that it effectively addresses the constraints of confounding factors and measurement errors that usually exist in observational settings, as the direction of causation is from the genetic variables toward the trait of interest, and not the reverse [17]. Thus, in this study, we aim to conduct a two-sample MR analysis (both univariable and multivariable MR analysis), which employs GWAS summary statistics from independent populations, to investigate the potential causal association between IBD (including CD and UC) and the risk of ASCVD (CAD and ischemic stroke), and provide more insights into the clinical management of IBD, and the prophylaxis of ASCVD.

## 2. Methods

### 2.1. Study Design

To perform a valid MR analysis, three fundamental assumptions need to be satisfied [18]. 

**Assumption** **1.***The relevance assumption, which requires the selected genetic variants, must be strongly associated with the exposure of interest*. 

**Assumption** **2.***The independence assumption indicates that no unmeasured confounders exist between the genetic variants and the outcome*. 

**Assumption** **3.***The exclusion restriction assumption requests that the effects of genetic variants on the outcome are only due to their effects on the exposure of interest (Figure 1A)*. 

A schematic overview of the study design is detailed in Figure 1B. Single-nucleotide polymorphisms (SNPs) associated with IBD, CD, and UC were retrieved from the latest genome-wide association study (GWAS) of European ancestry [16]. Outcome-associated SNPs were derived from other independent resources on coronary artery disease (CAD) [14] and ischemic stroke, with no overlapping [15]. Additionally, replication analyses were conducted, using outcome summary statistics from FinnGen consortium (R5 data release) [19] and UK Biobank (Neale Lab, www.nealelab.is/uk-biobank/, accessed on 30 May 2022). Furthermore, we included a positive and negative control outcome analysis of IBD, to evaluate the possible biases of horizontal pleiotropy and selective bias [20]. A positive control outcome is an outcome for which it is well established that the exposure is causally associated. In contrast, a negative outcome is an outcome for which the exposure is considered non-causal. The researchers responsible for all the GWASs included in our study had acquired informed consent and ethical approval from the participants and the relevant review committees. This study is reported according to the Strengthening the Reporting of Observational Studies in Epidemiology Using Mendelian Randomization (STROBE-MR) guidelines [21].

### 2.2. Selection of Genetic Variants for IBD

The GWAS summary statistics for IBD in individuals of European descent were derived from de Lange et al. [16], and comprise 25,042 IBD cases and 34,915 normal controls, including 12,194 cases and 28,072 controls for CD, and 12,366 cases and 33,609 controls for UC, respectively (Table 1). Diagnoses of IBD were confirmed via endoscopic, histopathological, and radiological approaches. These samples were genotyped and imputed to a combined reference panel of 4686 IBD patients and 6285 population controls from the UK IBD Genetics Consortium (UKIBDGC), UK10K project, and the 1000 Genomes Project (Phase 3 v5 release) [16]. We selected SNPs with a significant threshold of *p* < 5 × 10^−8^ and pruned SNPs in linkage disequilibrium with a strict r^2^ cut-off value of 0.001 (window size = 10,000 kb). Proxy variants with LD scores higher than 0.8 were accepted for instrumental variables lacking in outcome datasets. The F statistics for each SNP and phenotypic variance explained via genetic instruments were calculated, to avoid weak instrument bias (Appendix A). The statistical power was calculated through mRnd [22], a web-based tool for MR studies (https://shiny.cnsgenomics.com/mRnd/, accessed on 30 May 2022), where the significance level was set to 0.05 (Appendix A).

**Figure 1 biomedicines-11-02543-f001:**
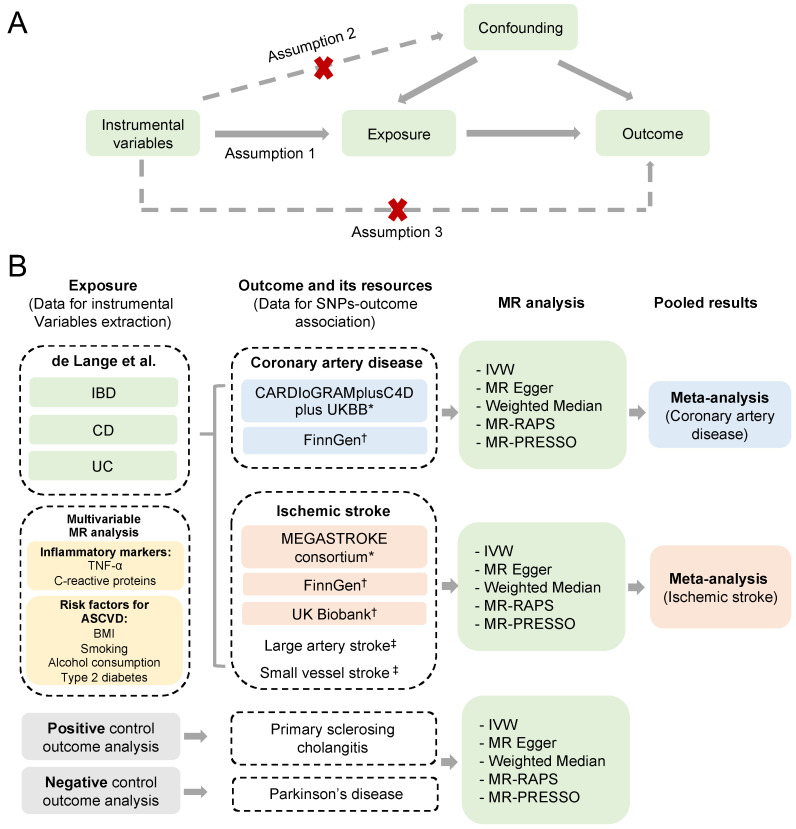
**Schematic overview of the study design.** (**A**) Illustration of the Mendelian randomization (MR) study and its three assumptions. (**B**) Diagram of our MR analyses using IBD, UC, and CD as exposures, and CAD and ischemic stroke from different resources as outcomes. MR analyses were performed for each outcome database, and were subsequently meta-analyzed to generate pooled estimates. Multivariable MR analyses were conducted via including the risk factors significantly associated with the ASCVD outcomes. Primary sclerosing cholangitis and Parkinson’s disease were used as positive and negative controls for the outcome analyses [14,15,16,19,23,24]. * The outcome data used in the main analysis. † The outcome data used in the replication analysis. ‡ The subgroup outcome data for ischemic stroke from the MEGASTROKE consortium. (IBD, inflammatory bowel disease; CD, Crohn’s disease; UC, ulcerative colitis; ASCVD, atherosclerotic cardiovascular disease; IVW, inverse-variance weighted method; MR-RAPS, MR robust adjusted profile score method; MR-PRESSO, MR pleiotropy residual sum and outlier method).

### 2.3. Summary Statistics for Atherosclerotic Cardiovascular Disease

The SNP–outcome associations for ASCVD were obtained from the following databases: CARDIoGRAMplusC4D consortium [14], MEGASTROKE consortium [15], FinnGen [19], and UK Biobank (detailed information regarding the datasets used can be found in Table 1). For coronary artery disease, summary statistics of the largest GWAS meta-analysis, containing 122,733 individuals diagnosed with CAD and 424,528 population controls, were obtained from CARDIoGRAMplusC4D consortium (the data include UK Biobank participants) [14]. For ischemic stroke and its subtypes, GWAS summary statistics were selected from the MEGASTROKE consortium with 34,217 cases and 406,111 controls of European ancestry [15]. Ischemic stroke and its two subtypes (large artery and small vessel) were classified according to the Trial of Org 10172 in Acute Stroke Treatment (TOAST) system [25]. Detailed information on this can be seen in previous studies [14,15,19].

### 2.4. Summary Statistics for Negative and Positive Control Outcomes

Previous evidence indicated a strong association between primary sclerosing cholangitis (PSC) and IBD, and suggested that a damaged gut barrier might be the possible underlying cause of PSC [26,27]. Thus, we utilized PSC as a positive control outcome. The genetic summary data for PSC were drawn from the International PSC Study Group, which contains 2187 cases and 12,019 population controls [23]. As for the negative control outcomes, the GWAS summary statistics for Parkinson’s disease were retrieved from the International Parkinson’s Disease Genomics Consortium (IPDGC) [24] (Table 1).

### 2.5. Statistical Analyses

In the primary analysis, we used the multiplicative random-effects inverse-variance weighted (IVW) method, which provides the greatest statistical power when all instruments are valid [20]. As the pleiotropy of SNPs universally exists, we further performed the pleiotropy-robust MR Egger, weighted median, MR robust adjusted profile score (MR-RAPS), and MR pleiotropy residual sum and outlier (MR-PRESSO) methods in sensitivity analyses. In general, the MR Egger and weighted median methods assumed that all and less than 50% of the genetic variants are invalid, respectively [28,29]. MR-RAPS provides robust estimation, to alleviate systematic and idiosyncratic pleiotropy, when there are many weak instruments [30]. MR-PRESSO could detect outlier SNPs, and adjust the horizontal pleiotropy by removing outliers [31]. The combined causal effect of IBD, CD, and UC on CAD and ischemic stroke were estimated using a fixed-effects meta-analysis. We also assessed the heterogeneity and directional pleiotropy using Cochran’s Q statistics and MR Egger intercept [32]. Leave-one-out analysis was performed, to identify a possible reliance on a particular variant, through excluding one SNP at a time for all valid SNPs in the IVW analysis. Moreover, we included the potential risk factors in the multivariable Mendelian randomization (MVMR) analysis, to further examine the validity of our study [33]. Of note, during the IV selection process of the inflammatory marker TNF-α, due to no instrumental variables passing the threshold *p* < 5 × 10^−8^, we relaxed the threshold to *p* < 5 × 10^−6^.

Given that all the exposure variables in the present study are binary, causal estimates were presented as an odds ratio with 95% confidential intervals (CIs), which can be interpreted as an average change in the outcome per 2.72-fold elevation in the prevalence of the corresponding exposure. The Bonferroni-corrected *p* = 0.002 (0.05/24) was considered as the significance threshold; the *p*-value < 0.05 was presented as the suggestive significance. All the analyses in this study were performed in R software (version 4.2.0), using the R packages “TwoSampleMR” (version: 0.5.6), “MRPRESSO” (version: 1.0), and “metafor” (version 3.4).

## 3. Results

### 3.1. Genetic Instruments

A total of 115 independent SNPs (*p* < 5 × 10^−8^) were identified for IBD, which explained 13.46% of the total phenotypic variation. The F statistics for each SNP ranged from 29.86 to 500.6 (median value = 70.33). Under the type I error, less than 0.05 assumption, there was ≥95% power to observe a causal association of an OR > 1.03 for CAD (Appendix A). As for CD and UC, 85 and 59 instrumental variables were retained, respectively, and the variance explained was 16.56% and 8.86% (Appendix A). Detailed information about all the SNPs is listed in Appendix A.

### 3.2. Main Analysis

Using the multiplicative random effects IVW method in the primary analysis, there was no evidence suggesting a causal association of genetical proxied IBD with CAD (OR 1.003, 95%CI 0.982 to 1.025) or any ischemic stroke (OR 0.994, 95%CI 0.97 to 1.018) (Table 2). Similarly, no causal association was identified for CD with CAD (OR 0.997, 95%CI 0.978 to 1.016), or ischemic stroke (OR 0.996, 95%CI 0.979 to 1.014), except for one subtype (large artery stroke, OR 1.044, 95%CI 1.003 to 1.087). Although the association between CD and large artery stroke reached the suggestive significant *p*-value (*p* = 0.037, Table 2), this significance should be interpreted with caution, given that, under the circumstance of an OR less than 1.05, the statistical power for CD on large artery stroke was lower than 27% (Appendix A). The results are similar for UC, for which we found no evidence of a causal relationship on the risk of CAD or ischemic stroke (Table 2).

Several pleiotropy-robust methods were then utilized in the sensitivity analysis (MR Egger, weighted median, MR-RAPS, and MR-PRESSO), and the findings were consistent with those of the primary analyses, indicating null causal association between IBD (including CD and UC, respectively) and CAD, ischemic stroke, or any subtypes of ischemic stroke, except for the case of CD contributing to large artery stroke, which still reached the suggestive significant threshold (Table 2).

### 3.3. Replication and Meta-Analysis

Additionally, we conducted replication analyses and a meta-analysis using ASCVD outcome summary statistics from FinnGen and UK Biobank, which further ensured the validity, reliability, and robustness of our study (Table 1). No causal evidence was found among the IBD (including CD and UC, respectively) and ASCVD outcomes. Only a decreased risk of ischemic stroke in IBD was detected via the MR Egger method. However, the associations were not detectable using MR-PRESSO with outliers removal or other methods (Table 3). In the meta-analysis (derived from the IVW method), the pooled ORs for CAD and ischemic stroke of a genetically predicted per-log-OR increase in IBD were 0.998 (95% CI: 0.982 to 1.014) and 1.000 (95% CI: 0.999 to 1.001), which support the above results of a null association between IBD and ASCVD, and between the subtypes of IBD (CD and UC) (Figure 2). The analyses of heterogeneity and directional pleiotropy are presented in Appendix A, with a moderate heterogeneity detected among several outcomes. No directional pleiotropy was detected using MR Egger intercept for all exposure–outcome analyses. Even the tests might be underpowered; however, when more robust methods were conducted to correct for directional pleiotropy, we observed similar estimates for ORs, indicating a non-causal association of IBD, CD, and UC with ASCVD. Besides, the results of our leave-one-out analyses remained consistent after removing one SNP at one time, which suggests that no single SNP dominated the results and strengthened the findings of our study.

### 3.4. Multivariable MR Analysis

Considering how IBD disease activity might influence the development of ASCVD, we further gathered some inflammatory factors (e.g., C-reactive protein and TNF-α) that reflect IBD disease activity [34,35], and some other common risk factors of ASCVD (e.g., body mass index [BMI], smoking initiation, alcohol consumption, and type 2 diabetes), into the univariate MR analysis, to identify potential factors causally associated with ASCVD. The dataset descriptions of inflammatory markers and the risk factors of ASCVD can be found in Appendix A. The MR estimates revealed that BMI and smoking initiation were significantly associated with an increased risk of CAD and ischemic stroke (Table 4). Thus, BMI and smoking initiation were included in the subsequent MVMR analysis. The MVMR approach could help us better understand the intricate interplay between the risk factors, and strengthen the validity of our study. After adjusting for BMI and smoking behavior, the results remained consistent, and no evidence of causal association was found between IBD (including CD and UC) and ASCVD (Table 5).

### 3.5. Positive and Negative Control Outcome Analyses

Additionally, to evaluate the possible biases from horizontal pleiotropy and selective bias, we applied PSC and Parkinson’s disease as the positive and negative control outcomes. As expected, the negative control outcome analysis revealed no causal effect of IBD on Parkinson’s disease, and the positive control outcome indicated a strong causal effect of IBD on the risk of PSC (Appendix A).

## 4. Discussion

In the present study, we did not observe a causal effect of IBD (including its two subtypes, CD and UC) on the risk of developing CAD and ischemic stroke. These results were further validated in replication, meta-, and sensitivity analysis with different pleiotropy-robust methods.

Previously, several large-scale observational studies suggested that patients with IBD may have an increased risk of developing ASCVD, including CAD and ischemic stroke. A Danish population-based study during 1997–2009, which consisted of over 4.5 million individuals, of whom 28,833 were IBD patients, indicated that patients with IBD had a significantly increased risk of ischemic heart disease, compared with the general population (incidence rate ratios = 1.59; 95% CI 1.50 to 1.69) [6]. A similar significant association was also identified in a nationwide French cohort study [36], in a population-based study conducted by Aarestrup et al. [37], and in a recently published retrospective multicenter cohort study from China [7]. However, previous evidence also indicated that IBD did not increase the hospitalization rate of acute myocardial infarction [38], and the mortality rate of cardiovascular disease (CVD) [39], suggesting that potential biases may confound the epidemiological findings. In addition, according to a recent meta-analysis [8], an increased risk of ischemic heart disease was mainly witnessed in women (OR, 1.26; 95%CI, 1.18–1.35) and population-based individuals (OR, 1.17; 95%CI, 1.11–1.23), whereas in men (OR, 1.05; 95%CI, 0.92–1.21) and hospital-based individuals (OR, 1.99; 95%CI, 0.51–7.78), this trend diminished. The results above indicate that potential biases might have been introduced in previous analyses. For example, procoagulant agents, such as oral contraceptives taken by females, may increase the risk of arterial thrombotic events in females. In addition, with more and frequent access to health professionals, compared with healthy individuals, IBD patients may be more likely to be diagnosed with CVD, which increases the risk of detection bias [40]. Given the above points, it is still inconclusive whether there is indeed a causal association between IBD and CAD and ischemic stroke.

Our study presents a different view of the association between IBD and ASCVD, via conducting a two-sample MR analysis. Surprisingly, our results differ from most observational studies, suggesting a null association between IBD and ASCVD. As observational studies are prone to unmeasured confounders, and it can be hard to determine the causal relationship, the observed association between IBD and ASCVD may be biased, and a causal association might not be reached. IBD may not directly correlate with ASCVD, but the medications taken by IBD patients are directly linked to ASCVD. Patients with a systemic usage of corticosteroids were found to be associated with a significantly increased risk of arterial thrombotic events (adjusted relative risk, 2.56; 95%CI, 2.18 to 2.99) [41,42], which indicates that corticosteroids might be the underlying cause of an increased ASCVD risk in IBD patients. Moreover, the effect of other medications, such as tumor necrosis factor inhibitors and 5-aminosalicylic acid (5-ASA) compounds, on ASCVD are still controversial, as some studies have suggested a protective role, while others have shown a reverse effect [6,43,44]. Given the complexity of multiple drug usage among IBD patients, and the still-unclear drug–drug interplay in the development of cardiovascular atherosclerosis among IBD patients, the findings of previous observational studies might be biased by these medications, and we may need more well-designed studies to confirm the influence of IBD-specific drugs on ASCVD. Apart from this, the multicenter randomized control (CANTOS) trial by Ridker et al. demonstrated that an anti-inflammatory drug, canakinumab, which targets cytokines IL-1 and IL-6, could decrease ASCVD events [45,46]. Thus, it still cannot be ruled out that, instead of the diagnosis of IBD, the accompanied inflammatory mediators, such as IL-1 and IL-6, drove the increased risk of ASCVD [47].

To summarize, there are several strengths to our study. Firstly, we utilized integrated approaches in the assessment of the causal effect of IBD and two of its subtypes, CD and UC, on the risk of ASCVD. Compared with conventional epidemiological studies, the two-sample MR analysis approach provides a higher level of causal association evidence, due to it being less susceptible to potential biases and confounders when the primary assumptions are met [12]. We have also gathered the latest and largest GWAS summary statistics for IBD and ASCVD outcomes from different consortiums in the main MR analysis, to make our results plausible. Secondly, we conducted a replication analysis using data from two large cohort studies, UK Biobank and FinnGen [19], to validate the primary analysis. According to multiple pleiotropy-robust methods in the sensitivity analysis, the effect estimations from different data sources were consistent. Moreover, the results of the meta-analyses and MVMR analysis further strengthened our findings. In conclusion, we included different data resources, multiple statistical approaches, a meta-analysis, and MVMR analysis in our study, to assess the causal association of IBD with ASCVD. All these methods generated consistent results, and suggest that IBD-specific pharmacological interventions may not be required for the primary prevention of ASCVD in patients with IBD.

However, some limitations need to be considered. Firstly, the GWAS summary statistics used in our study were mainly derived from the European population, limiting the generalization of our findings to other ethnic populations. Secondly, we only considered IBD as a dichotomous variable, rather than a broad IBD disease spectrum with multiple disease statuses, while disease severity and activity in IBD were reported to be associated with the risk of ASCVD [36]. Although we did not find that inflammatory markers such as CRP and TNF-α are causally associated with the risk of ASCVD, still, the influence of inflammatory factors on ASCVD cannot be ruled out, as limited valid instrumental variables were available in the study (e.g., TNF-α). Moreover, IBD is characterized by its fluctuating disease course; due to the lack of direct GWAS data on disease activity and severity in IBD, we were unable to conduct an MR analysis to directly assess the associations between IBD activity and ASCVD. Of note, the genetic characteristics of IBD disease activity might be distinct from the IBD incidence. In such a case, disease flare-ups might still predispose IBD patients to ASCVD. Thirdly, subgroup analyses, such as different medication subgroup analyses, were not available through the currently released data. Thus, future analysis should be well designed, and carefully analyzed, via subgrouping IBD patients with different medication schemes.

## 5. Conclusions

Our MR analysis revealed no evidence to support a causal association between IBD (including its two subgroups, CD and UC) and ASCVD (CAD and ischemic stroke). Previous results from observational studies might be biased via uncontrolled confoundings (such as IBD-specific medications and detection bias, etc.). Further research is needed, to clarify the risk factors that causally affect arterial thrombotic events.

## Figures and Tables

**Figure 2 biomedicines-11-02543-f002:**
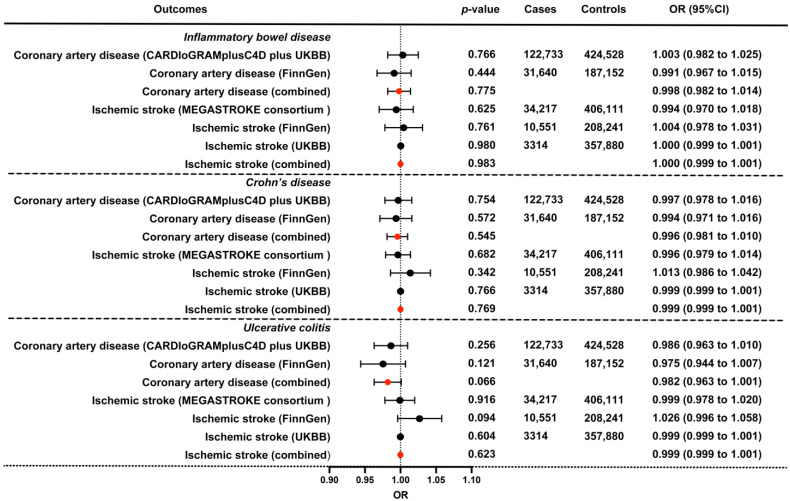
The combined IVW causal effect estimation from a fixed effect meta-analysis for the genetical proxied IBD on the risk of CAD and ischemic stroke. Estimated ORs were obtained from different datasets, and combined using a fixed-effect meta-analysis model, which represents the average change in the outcome per 2.72-fold elevation in the prevalence of the corresponding exposure. The bars indicate 95% confidential intervals, and the red dots indicate the combined effect. UKBB, UK Biobank; OR, odds ratio; 95%CI, 95% confidence interval.

**Table 1 biomedicines-11-02543-t001:** Description of datasets used for analysis.

Phenotype	Data Sources	No. of Cases	No. of Controls	Population
**Exposures**				
Inflammatory bowel disease	de Lange et al. [16]	25,042	34,915	European
Crohn’s disease	de Lange et al. [16]	12,194	28,072	European
Ulcerative colitis	de Lange et al. [16]	12,366	33,609	European
**Outcomes**				
Coronary artery disease	CARDIoGRAMplusC4D plus UKBB * [14]	122,733	424,528	Majority European
	FinnGen † [19]	31,640	187,152	European
Ischemic stroke	MEGASTROKE consortium * [15]	34,217	406,111	European
	FinnGen †	10,551	208,241	European
	UKBB †	3314	357,880	European
Large artery stroke	MEGASTROKE consortium *	4373	406,111	European
Small vessel stroke	MEGASTROKE consortium *	5386	192,662	European
**Positive and negative control outcome**				
Primary sclerosing cholangitis	International PSC Study Group [23]	2187	12,019	Majority European
Parkinson’s disease	International Parkinson’s Disease Genomics Consortium [24]	33,674	449,056	European

* The outcome data used in the main analysis. † The outcome data used in the replication analysis. UKBB, UK Biobank.

**Table 2 biomedicines-11-02543-t002:** Mendelian randomization (MR) estimates for the causal effect of genetically proxied IBD, CD, and UC on coronary artery disease and ischemic stroke (including large artery stroke and small vessel stroke).

Exposure	Outcome	IVW		MR Egger		Weighted Median		MR-RAPS		MR-PRESSO	
		OR (95%CI)	*p*-Value	OR (95%CI)	*p*-Value	OR (95%CI)	*p*-Value	OR (95%CI)	*p*-Value	OR (95%CI)	*p*-Value
IBD	Coronary artery disease	1.003 (0.982 to 1.025)	0.766	0.962 (0.908 to 1.018)	0.181	1.002 (0.979 to 1.027)	0.839	1.001 (0.981 to 1.020)	0.943	1.002 (0.988 to 1.016)	0.806
	Ischemic stroke *	0.994 (0.97 to 1.018)	0.625	0.969 (0.913 to 1.029)	0.312	0.983 (0.952 to 1.016)	0.314	0.988 (0.966 to 1.010)	0.290	0.992 (0.973 to 1.01)	0.390
	Ischemic stroke * (large artery atherosclerosis)	1.013 (0.955 to 1.074)	0.668	1.047 (0.904 to 1.212)	0.542	1.05 (0.972 to 1.135)	0.215	1.025 (0.967 to 1.086)	0.408	1.018 (0.973 to 1.065)	0.443
	Ischemic stroke * (small-vessel)	1.021 (0.976 to 1.067)	0.369	0.937 (0.84 to 1.046)	0.251	0.974 (0.906 to 1.048)	0.481	1.011 (0.967 to 1.057)	0.638	1.026 (0.987 to 1.066)	0.204
CD	Coronary artery disease	0.997 (0.978 to 1.016)	0.754	0.976 (0.926 to 1.028)	0.362	0.996 (0.978 to 1.015)	0.702	0.998 (0.980 to 1.016)	0.835	0.996 (0.983 to 1.008)	0.474
	Ischemic stroke *	0.996 (0.979 to 1.014)	0.682	0.959 (0.916 to 1.004)	0.075	1.003 (0.976 to 1.031)	0.805	0.996 (0.977 to 1.014)	0.651	0.998 (0.982 to 1.015)	0.852
	Ischemic stroke * (large artery atherosclerosis)	1.044 (1.003 to 1.087)	0.037	1.037 (0.932 to 1.155)	0.507	1.08 (1.013 to 1.151)	0.018	1.05 (1.004 to 1.097)	0.032	1.041 (1.001 to 1.082)	0.046
	Ischemic stroke * (small-vessel)	1.012 (0.979 to 1.047)	0.477	0.908 (0.825 to 0.999)	0.050	0.995 (0.942 to 1.052)	0.869	1.014 (0.977 to 1.053)	0.459	1.015 (0.983 to 1.049)	0.367
UC	Coronary artery disease	0.986 (0.963 to 1.010)	0.256	0.967 (0.902 to 1.037)	0.355	0.99 (0.968 to 1.014)	0.416	0.988 (0.967 to 1.01)	0.275	0.994 (0.979 to 1.009)	0.414
	Ischemic stroke *	0.999 (0.978 to 1.020)	0.916	0.968 (0.909 to 1.03)	0.309	0.982 (0.951 to 1.015)	0.289	0.997 (0.975 to 1.02)	0.808	1.001 (0.983 to 1.02)	0.903
	Ischemic stroke * (large artery atherosclerosis)	1.036 (0.974 to 1.101)	0.263	1.018 (0.848 to 1.223)	0.847	1.086 (0.998 to 1.181)	0.054	1.054 (0.991 to 1.122)	0.096	1.053 (1 to 1.109)	0.055
	Ischemic stroke * (small-vessel)	1.033 (0.98 to 1.090)	0.224	0.963 (0.823 to 1.128)	0.645	1.006 (0.936 to 1.081)	0.876	1.028 (0.974 to 1.085)	0.314	1.021 (0.973 to 1.072)	0.400

IBD, inflammatory bowel disease; CD, Crohn’s disease; UC, ulcerative colitis; IVW, inverse-variance weighted method; MR-RAPS, MR robust adjusted profile score method; MR-PRESSO, MR pleiotropy residual sum and outlier method; OR, odds ratio; 95%CI, 95% confidential interval; * data from MEGASTROKE consortium.

**Table 3 biomedicines-11-02543-t003:** The replication analysis for a causal relationship of IBD, CD, and UC with coronary heart disease and ischemic stroke from FinnGen and UK Biobank.

Exposure	Outcome	IVW		MR Egger		Weighted Median		MR-RAPS		MR-PRESSO	
		OR (95%CI)	*p*-Value	OR (95%CI)	*p*-Value	OR (95%CI)	*p*-Value	OR (95%CI)	*p*-Value	OR (95%CI)	*p*-Value
IBD	Coronary artery disease (FinnGen)	0.991 (0.967 to 1.015)	0.444	0.987 (0.947 to 1.030)	0.551	0.980 (0.947 to 1.014)	0.251	0.988 (0.965 to 1.011)	0.300	0.987 (0.968 to 1.007)	0.217
	Ischemic stroke (FinnGen)	1.004 (0.978 to 1.031)	0.761	1.028 (0.982 to 1.076)	0.233	1.021 (0.976 to 1.068)	0.361	1.006 (0.979 to 1.033)	0.659	0.979 (0.953 to 1.005)	0.111
	Ischemic stroke (UKBB)	1.000 (0.999 to 1.001)	0.98	0.998 (0.997 to 0.999)	0.026	0.999 (0.999 to 1.001)	0.393	0.999 (0.999 to 1.001)	0.938	0.999 (0.999 to 1.001)	0.935
CD	Coronary artery disease (FinnGen)	0.994 (0.971 to 1.016)	0.572	0.972 (0.915 to 1.033)	0.362	1.001 (0.974 to 1.029)	0.925	0.989 (0.965 to 1.012)	0.344	0.990 (0.970 to 1.010)	0.333
	Ischemic stroke (FinnGen)	1.013 (0.986 to 1.042)	0.342	0.993 (0.922 to 1.070)	0.862	1.010 (0.972 to 1.050)	0.602	1.015 (0.987 to 1.043)	0.311	0.976 (0.950 to 1.002)	0.076
	Ischemic stroke (UKBB)	0.999 (0.999 to 1.001)	0.766	0.999 (0.998 to 1.001)	0.226	1.000 (0.999 to 1.001)	0.524	0.999 (0.999 to 1.001)	0.721	0.999 (0.999 to 1.001)	0.690
UC	Coronary artery disease (FinnGen)	0.975 (0.944 to 1.007)	0.121	0.945 (0.844 to 1.057)	0.317	0.973 (0.938 to 1.010)	0.157	0.979 (0.949 to 1.010)	0.186	0.986 (0.961 to 1.012)	0.304
	Ischemic stroke (FinnGen)	1.026 (0.996 to 1.058)	0.094	0.935 (0.852 to 1.025)	0.16	1.008 (0.962 to 1.056)	0.742	1.022 (0.989 to 1.055)	0.195	1.005 (0.971 to 1.040)	0.785
	Ischemic stroke (UKBB)	0.999 (0.999 to 1.001)	0.604	0.998 (0.997 to 1.001)	0.155	0.999 (0.998 to 1.001)	0.251	0.999 (0.999 to 1.001)	0.611	0.999 (0.999 to 1.001)	0.927

IBD, inflammatory bowel disease; CD, Crohn’s disease; UC, ulcerative colitis; UKBB, UK Biobank; IVW, inverse-variance weighted method; MR-RAPS, MR robust adjusted profile score method; MR-PRESSO, MR pleiotropy residual sum and outlier method; OR, odds ratio; 95%CI, 95% confidential interval.

**Table 4 biomedicines-11-02543-t004:** Univariate MR estimates between inflammatory markers, common risk factors, and ASCVD.

Exposure	Outcome	Method	OR (95%CI)	*p*-Value
TNF-α	Coronary artery disease (CARDIoGRAMplusC4D plus UKBB)	IVW	0.993 (0.941 to 1.047)	0.787
	Coronary artery disease (FinnGen)	IVW	1.225 (0.913 to 1.644)	0.176
	Ischemic stroke (MEGASTROKE)	IVW	1.000 (0.999 to 1.001)	0.927
	Ischemic stroke (large artery atherosclerosis) (MEGASTROKE)	IVW	0.987 (0.943 to 1.032)	0.559
	Ischemic stroke (small vessel) (MEGASTROKE)	IVW	0.945 (0.864 to 1.033)	0.214
	Ischemic stroke (FinnGen)	IVW	1.026 (0.835 to 1.261)	0.809
	Ischemic stroke (UKBB)	IVW	1.056 (0.790 to 1.411)	0.715
CRP	Coronary artery disease (CARDIoGRAMplusC4D plus UKBB)	IVW	0.908 (0.828 to 0.996)	0.041
	Coronary artery disease (FinnGen)	IVW	0.969 (0.869 to 1.079)	0.563
	Ischemic stroke (MEGASTROKE)	IVW	1.024 (0.957 to 1.096)	0.486
	Ischemic stroke (large artery atherosclerosis) (MEGASTROKE)	IVW	0.999 (0.998 to 1.000)	0.150
	Ischemic stroke (small vessel) (MEGASTROKE)	IVW	1.09 (0.920 to 1.291)	0.321
	Ischemic stroke (FinnGen)	IVW	0.989 (0.848 to 1.154)	0.889
	Ischemic stroke (UKBB)	IVW	0.984 (0.914 to 1.060)	0.678
**BMI**	Coronary artery disease (CARDIoGRAMplusC4D plus UKBB)	IVW	1.507 (1.406 to 1.615)	3.20 × 10^−31^
	Coronary artery disease (FinnGen)	IVW	1.002 (1.001 to 1.004)	0.004
	Ischemic stroke (MEGASTROKE)	IVW	1.314 (1.101 to 1.568)	0.003
	Ischemic stroke (large artery atherosclerosis) (MEGASTROKE)	IVW	1.337 (1.232 to 1.451)	3.71 × 10^−12^
	Ischemic stroke (small vessel) (MEGASTROKE)	IVW	1.162 (1.077 to 1.254)	1.10 × 10^−4^
	Ischemic stroke (FinnGen)	IVW	1.120 (0.954 to 1.315)	0.167
	Ischemic stroke (UKBB)	IVW	1.155 (1.047 to 1.275)	0.004
Alcoholic drinks per week	Coronary artery disease (CARDIoGRAMplusC4D plus UKBB)	IVW	1.273 (0.972 to 1.667)	0.079
	Coronary artery disease (FinnGen)	IVW	0.799 (0.522 to 1.223)	0.302
	Ischemic stroke (MEGASTROKE)	IVW	1.124 (0.901 to 1.402)	0.300
	Ischemic stroke (large artery atherosclerosis) (MEGASTROKE)	IVW	1.606 (0.944 to 2.734)	0.081
	Ischemic stroke (small vessel) (MEGASTROKE)	IVW	0.713 (0.411 to 1.237)	0.229
	Ischemic stroke (FinnGen)	IVW	1.136 (0.812 to 1.591)	0.456
	Ischemic stroke (UKBB)	IVW	1.003 (0.997 to 1.008)	0.326
**Smoking initiation**	Coronary artery disease (CARDIoGRAMplusC4D plus UKBB)	IVW	1.242 (1.127 to 1.368)	1.15 × 10^−5^
	Coronary artery disease (FinnGen)	IVW	1.181 (1.007 to 1.385)	0.041
	Ischemic stroke (MEGASTROKE)	IVW	1.137 (1.027 to 1.258)	0.013
	Ischemic stroke (large artery atherosclerosis) (MEGASTROKE)	IVW	1.575 (1.283 to 1.935)	1.47 × 10^−5^
	Ischemic stroke (small vessel) (MEGASTROKE)	IVW	1.219 (1.093 to 1.36)	3.63 × 10^−4^
	Ischemic stroke (FinnGen)	IVW	1.195 (0.941 to 1.517)	0.145
	Ischemic stroke (UKBB)	IVW	1.002 (1.000 to 1.004)	0.067
Type 2 diabetes	Coronary artery disease (CARDIoGRAMplusC4D plus UKBB)	IVW	1.014 (0.998 to 1.03)	0.090
	Coronary artery disease (FinnGen)	IVW	1.009 (0.948 to 1.074)	0.767
	Ischemic stroke (MEGASTROKE)	IVW	0.997 (0.981 to 1.014)	0.729
	Ischemic stroke (large artery atherosclerosis) (MEGASTROKE)	IVW	1.000 (1.000 to 1.000)	0.290
	Ischemic stroke (small vessel) (MEGASTROKE)	IVW	0.996 (0.986 to 1.005)	0.397
	Ischemic stroke (FinnGen)	IVW	1.001 (0.983 to 1.020)	0.895
	Ischemic stroke (UKBB)	IVW	1.003 (0.970 to 1.038)	0.853

TNF-α, tumor necrosis factor alpha; CRP, C-reactive protein; IVW, inverse-variance weighted method; UKBB, UK Biobank; ASCVD, atherosclerotic cardiovascular disease; OR, odds ratio; 95%CI, 95% confidential interval.

**Table 5 biomedicines-11-02543-t005:** Multivariable MR estimates for the causal effect of genetical proxied IBD, CD, UC (adjusted for BMI and smoking initiation), and ASCVD.

Exposure	Outcome	Method	OR (95%CI)	*p*-Value
IBD, adjusted for BMI and smoking initiation	Coronary artery disease (CARDIoGRAMplusC4D plus UKBB)	IVW	1.009 (0.982 to 1.038)	0.507
	Coronary artery disease (FinnGen)	IVW	0.998 (0.964 to 1.033)	0.921
	Ischemic stroke (MEGASTROKE)	IVW	1.000 (0.969 to 1.032)	0.991
	Ischemic stroke (large artery atherosclerosis) (MEGASTROKE)	IVW	0.993 (0.922 to 1.069)	0.847
	Ischemic stroke (small vessel) (MEGASTROKE)	IVW	1.014 (0.948 to 1.085)	0.688
	Ischemic stroke (FinnGen)	IVW	0.98 (0.941 to 1.02)	0.322
	Ischemic stroke (UKBB)	IVW	1.000 (1.000 to 1.001)	0.590
CD, adjusted for BMI and smoking initiation	Coronary artery disease (CARDIoGRAMplusC4D plus UKBB)	IVW	1.009 (0.988 to 1.030)	0.410
	Coronary artery disease (FinnGen)	IVW	1.015 (0.989 to 1.042)	0.267
	Ischemic stroke (MEGASTROKE)	IVW	1.002 (0.979 to 1.026)	0.844
	Ischemic stroke (large artery atherosclerosis) (MEGASTROKE)	IVW	1.043 (0.988 to 1.101)	0.128
	Ischemic stroke (small vessel) (MEGASTROKE)	IVW	1.025 (0.975 to 1.078)	0.339
	Ischemic stroke (FinnGen)	IVW	0.996 (0.965 to 1.028)	0.792
	Ischemic stroke (UKBB)	IVW	1.000 (1.000 to 1.001)	0.846
UC, adjusted for BMI and smoking initiation	Coronary artery disease (CARDIoGRAMplusC4D plus UKBB)	IVW	1.013 (0.985 to 1.040)	0.367
	Coronary artery disease (FinnGen)	IVW	1.017 (0.982 to 1.053)	0.343
	Ischemic stroke (MEGASTROKE)	IVW	1.010 (0.979 to 1.042)	0.537
	Ischemic stroke (large artery atherosclerosis) (MEGASTROKE)	IVW	0.994 (0.923 to 1.071)	0.879
	Ischemic stroke (small vessel) (MEGASTROKE)	IVW	1.040 (0.971 to 1.113)	0.266
	Ischemic stroke (FinnGen)	IVW	1.019 (0.979 to 1.061)	0.362
	Ischemic stroke (UKBB)	IVW	1.000 (0.999 to 1.001)	0.859

IBD, inflammatory bowel disease; CD, Crohn’s disease; UC, ulcerative colitis; UKBB, UK Biobank; IVW, inverse variance weighted method; BMI, body mass index; ASCVD, atherosclerotic cardiovascular disease; OR, odds ratio; 95%CI, 95% confidential interval.

## Data Availability

This study was conducted using publicly available data. The GWAS summary statistics for IBD, CD, and CD are available at ftp://ftp.sanger.ac.uk/pub/project/humgen/summary_statistics/human/2016-11-07/, accessed on 30 May 2022. The CAD meta-GWAS summary data for CARDIoGRAMplusC4D and UK Biobank could be downloaded from http://www.cardiogramplusc4d.org/data-downloads/, accessed on 30 May 2022. The GWAS summary statistics of ischemic stroke and its subtypes are available at https://www.megastroke.org/, accessed on 30 May 2022. The GWAS summary data of CAD and ischemic stroke from FinnGen and UK Biobank are available at https://finngen.gitbook.io/documentation/v/r5/ and http://www.nealelab.is/uk-biobank, respectively, accessed on 30 May 2022. The GWAS summary statistics for TNF-α are available at http://computationalmedicine.fi/data#Cytokine_GWAS, accessed on 28 August 2023. The GWAS summary statistics for smoking initiation and alcohol consumption are available at https://genome.psych.umn.edu/index.php/GSCAN, accessed on 28 August 2023. The GWAS summary statistics for BMI are available at https://portals.broadinstitute.org/collaboration/giant/index.php/GIANT_consortium_data_files, accessed on 28 August 2023. We retrieved the above GWAS summary statistics through the IEU OpenGWAS project (https://gwas.mrcieu.ac.uk/, accessed on 30 May 2022 and 28 August 2023).

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
