# Peer review of "Evaluating the Causal Association between Inflammatory Bowel Disease and Risk of Atherosclerotic Cardiovascular Disease: Univariable and Multivariable Mendelian Randomization Study"

_biomedicines, 2023, doi:10.3390/biomedicines11092543_

Round 1

Reviewer 1 Report

Review for the manuscript

Appraising the causal association between inflammatory bowel disease and risk of atherosclerotic cardiovascular disease: perspectives from a two-sample Mendelian randomization studysubmitted to Biomedicines.

Dear Editor, thank you for the invitation to review this exciting manuscript. However, I have few suggestions before it can be considered for publication.

Overall comments: This is an interesting study that aimed to evaluate the potential effect of Inflammatory Bowel Diseases on the risk of coronary artery disease and ischemic stroke using two-sample Mendelian randomization study.

ABSTRACT

It is adequately performed.

I suggest including Crohn´s disease and ulcerative colitis among the keywords.

INTRODUCTION

        I suggest including newer references in the first paragraph. Many nice studies published in 2022 and 2023 can be found in PUBMED.

        Figure 1 is really very good and elucidative. However, it should be in the Methods section instead of Introduction.

METHODS

This section is adequate.

RESULTS

        The title of Table 2 is “Table 2. Mendelian randomization (MR) estimates for the causal effect of genetical proxied IBD, CD, UC on coronary artery disease, ischemic stroke (indluding large artery stroke, small vessel stroke and cardioembolic stroke).”. Please, modify for “Table 2. Mendelian randomization (MR) estimates for the causal effect of genetical proxied IBD, CD, UC on coronary artery disease, ischemic stroke (including large artery stroke, small vessel stroke, and cardioembolic stroke).”

CONCLUSION

        This section is adequately described.

REFERENCES

        As pointed out, I suggest including newer references in the Introduction section. 

Reviewer 2 Report

In the present original article Liu et al used a Mendelian randomization model to assess whether IBD patients may have an increased risk of atherosclerotic cardiovascular disease. They did not find a causal effect between IBD and ischemic stroke/ cardiovascular artery disease. The paper has several drawbacks in my opinion:

1) Genetic basis of IBD has a too low impact on the etiopathogenesis in order to set a Mendelian study. Environmental factors are much more important in IBD.

2) Disease activity was not evaluated. This variable is indeed one of the most relevant ones contributing to cardiovascular disease.

3) Furthermore, the analysis was not adjusted for several other factors such as age or IBD-specific drugs.

4) I have some doubts about using Parkinson disease as negative control, as such patients are much older than IBD.

Reviewer 3 Report

Dera Editor, 

I had a great pleasure to be able to read an article entitled: "Appraising the causal association between inflammatory bowel disease and risk of atherosclerotic cardiovascular disease: perspectives from two-sample Mendelian randomization study", in which the authors aimed to examine the potential effect of IBD, CD and UC on the risk of CAD and ischemic stroke using two-sample Mendelian randomization. 

The difference, between observational clinical studies and presented study could be, at least in part explained by gene expression. 

Patients from observational studies may be more often less controlled, with more exacerbations (eg. hospitalizations, outpatients visits) and more severe baseline/average inflammation level compared to the current group. This examination should be supplemented with, for example, anti-inflammatory or immunosuppressive treatment and the average concentration of pro-inflammatory markers.

Also statistical analysis should be adjusted for these factors. 

Reviewer 4 Report

The study under review, investigating through two-sample Mendelian randomization the causal link between inflammatory bowel disease and the risk of atherosclerotic cardiovascular outcomes, concluded that the higher incidence of coronary artery disease or stroke in IBD patients, reported previously by a multitude of large scale observational studies, is not consistent. Though, accelerated atherosclerosis due to increased inflammation or pro-atherogenic effects of IBD therapy could explain the clinical findings.

Comments

Abstract, conclusions – would be useful to briefly underline the specific characteristics of MR study versus the many observational studies that suggested a different outcome.

Introduction

Line 52 – please elaborate briefly on the ‘potential confounding and unmeas- 51 ured biases’

Line 55 ‘…provide a valuable opportunity to explore the internal causal effects’ – a more relevant formulation for the ‘internal causal effects’ would be recommended here, in the introduction

The purpose of the introduction is, among others, to facilitate the understanding of the manuscript for MDs or biologists that are not very familiar with the MR studies. Therefore, their advantages over observational research should be underlined in an accessible manner.

Lines 57-64 – our recommendation is to summarize this paragraph in an informative manner, for readers who are not very familiar to MR; a brief version of the present structure can be included in the Methods.

Line 65 – please define here the ‘two-sample MR analysis’

Figure 1: the abbreviations should be defined in the figure legend

Methods

Line 92: ‘ethical approval from participants and relevant review committees.’ – the ethic committees should be clearly identified here

Figure 1B and Table 1: Cardioembolic stroke is not related to atherosclerosis; therefore, seems to be an inappropriate outcome in your study

Results

Line 186: would be useful for the reader to provide a brief explanation of why you are conducting replication analyses

Line 201: could you elaborate here on the significance of the results of leave-one-out analyses?

Discussions

Line 231: some example of potential biases would be welcomed

Line 242: ‘According to our analysis, the results do not support a causal relationship between IBD and CAD and ischemic stroke’ is redundant

Some typos were detected

Round 2

Reviewer 2 Report

Answers did not solved the criticisms raised by the review, in particular points 1-3.

none

Author Response

We have updated our reply to Q1-Q3 of reviewer 2 and made a major revision in our manuscript. Please refer to the attachment file below.

Reviewer 4 Report

The rationale behind including cardioembolic stroke as an outcome measure (lines 139- 141) is not accurate, to our opinion, and therefore not acceptable

Author Response

Thanks for the reviewer's comment, we understand the reviewer's concern and we have removed the cardioembolic stroke in the subtype analysis of ischemic stroke, moreover, we have added multivariable analysis in our study (adjusted for potential risk factors), please refer to the revised manuscript.

Round 3

Reviewer 2 Report

no comment

none

Reviewer 4 Report

The comments were addressed adequately